# Adenosinergic System and BDNF Signaling Changes as a Cross-Sectional Feature of RTT: Characterization of *Mecp2* Heterozygous Mouse Females

**DOI:** 10.3390/ijms242216249

**Published:** 2023-11-13

**Authors:** Catarina Miranda-Lourenço, Jéssica Rosa, Nádia Rei, Rita F. Belo, Ana Luísa Lopes, Diogo Silva, Cátia Vieira, Teresa Magalhães-Cardoso, Ricardo Viais, Paulo Correia-de-Sá, Ana M. Sebastião, Maria J. Diógenes

**Affiliations:** 1Instituto de Farmacologia e Neurociências, Faculdade de Medicina, Universidade de Lisboa, 1649-028 Lisboa, Portugal; catarinalourenco@medicina.ulisboa.pt (C.M.-L.); anaseb@medicina.ulisboa.pt (A.M.S.); 2Instituto de Medicina Molecular João Lobo Antunes, Faculdade de Medicina, Universidade de Lisboa, 1649-028 Lisboa, Portugal; 3Laboratório de Farmacologia e Neurobiologia/MedInUP, Instituto de Ciências Biomédicas Abel Salazar—Universidade do Porto (ICBAS-UP), 4050-313 Porto, Portugaldiogo.a.r.silva@hotmail.com (D.S.); farmacol@icbas.up.pt (P.C.-d.-S.)

**Keywords:** Rett syndrome, *Mecp2* heterozygous females, adenosine, BDNF

## Abstract

Rett Syndrome is an X-linked neurodevelopmental disorder (RTT; OMIM#312750) associated to *MECP2* mutations. MeCP2 dysfunction is seen as one cause for the deficiencies found in brain-derived neurotrophic factor (BDNF) signaling, since BDNF is one of the genes under MeCP2 jurisdiction. BDNF signaling is also dependent on the proper function of the adenosinergic system. Indeed, both BDNF signaling and the adenosinergic system are altered in *Mecp2*-null mice (*Mecp2*^−/y^), a representative model of severe manifestation of RTT. Considering that symptoms severity largely differs among RTT patients, we set out to investigate the BDNF and ADO signaling modifications in *Mecp2* heterozygous female mice (*Mecp2*^+/−^) presenting a less severe phenotype. Symptomatic *Mecp2*^+/−^ mice have lower BDNF levels in the cortex and hippocampus. This is accompanied by a loss of BDNF-induced facilitation of hippocampal long-term potentiation (LTP), which could be restored upon selective activation of adenosine A_2A_ receptors (A_2A_R). While no differences were observed in the amount of adenosine in the cortex and hippocampus of *Mecp2*^+/−^ mice compared with healthy littermates, the density of the A_1_R and A_2A_R subtype receptors was, respectively, upregulated and downregulated in the hippocampus. Data suggest that significant changes in BDNF and adenosine signaling pathways are present in an RTT model with a milder disease phenotype: *Mecp2*^+/−^ female animals. These features strengthen the theory that boosting adenosinergic activity may be a valid therapeutic strategy for RTT patients, regardless of their genetic penetrance.

## 1. Introduction

Rett Syndrome (RTT; OMIM#312750) is a neurodevelopmental syndrome that affects mostly girls. In the last 20 years, it has presented a prevalence of 5–10:100.000 cases [1,2]. RTT patients are clinically described as presenting an apparently normal development, reaching most expected development milestones until 6–18 months after birth, when some signs and symptoms of the disease become apparent [3,4,5]. As the disease progresses, the following symptoms may arise: stereotypic hand movement and loss of purposeful hand movements, motor coordination features, epileptic seizures, language and learning deficits, and cognitive impairments ranging from mild to severe [6].

The vast majority of RTT cases are caused by mutations in the Methyl-CpG-binding protein 2 (*MECP2*) gene, present in the X chromosome [1]. *MECP2* encodes the homonymous protein MeCP2, which displays a wide range of functions, including a critical role in the regulation of genetic transcription [7,8]. One of the genes under MeCP2 control is the brain-derived neurotrophic factor (*BDNF*) gene [9]. BDNF is an important neurotrophin, with impactful functions during neuronal development and maturation but also in the modulation of synaptic plasticity of mature neuronal circuits [10] via the activation of the high-affinity tropomyosin receptor kinase B full-length (TrkB-FL) receptors [11,12]. In addition, BDNF can also bind, though with less affinity, to TrkB truncated isoforms (TrkB-Tc), which act as a negative effectors of TrkB-FL receptors [13,14].

Previous findings from our group and others have shown that the brain of *Mecp2*^−/y^ male mice possess lower levels of the BDNF protein [15,16,17,18]. Moreover, there is also a reduction in the levels of TrkB-FL receptor in the cortex and hippocampus of *Mecp2*^−/y^ male mice, which may also contribute to BDNF-associated functional deficits found in this RTT model [15]. These findings strengthen the theory that BDNF signaling is downregulated in *Mecp2*^−/y^ animals and may be paramount to RTT pathophysiology. On top of that, we also showed that extracts of the hippocampus and cortex from symptomatic *Mecp2*^−/y^ mice display lower levels of adenosine and its membrane receptor A_2A_R when compared with wild-type (WT) littermates. 

Mounting evidence indicates that synergism between adenosine A_2A_R and BDNF receptors is crucial for a suitable neuronal function and synaptic transmission. This includes the maintenance of adequate levels of BDNF and TrkB-FL receptors and consequently its mediated actions, such as the preservation of long-term potentiation (LTP), considered an electrophysiological model for the basic mechanisms involved in learning and memory formation [19,20,21,22,23,24,25,26,27]. Impairments in the A_2A_R/BDNF crosstalk were observed in symptomatic *Mecp2*^−/y^ mice. And it was proved, for the first time, that some of BDNF functional deficits in *Mecp2*^−/y^ mice can be overcome by selective A_2A_R activation, leading us to hypothesize that boosting BDNF signaling by increasing the availability of endogenous adenosine and/or the activity of A_2A_R could be a useful therapeutic strategy for RTT patients [15].

Although the use of symptomatic RTT *Mecp2*-null animals in research has the advantage of reproducibility and relatively low variability in the results produced [28,29], RTT is a disease characterized by a broad spectrum of clinical manifestations with variable severity, as predicted by the existence of different patterns of MeCP2 expression levels [30,31,32]. This prompted the need for investigating *Mecp2* heterozygous female mice (*Mecp2*^+/−^) presenting variable phenotypes, which are less severe when compared with *Mecp2*-null male mice due to X-chromosome inactivation (XCI) as a proxy of the majority of RTT female clinical patients [33]. Herein, we set out to investigate alterations in adenosinergic/BDNF signaling in the cortex and hippocampus of *Mecp2*^+/−^ heterozygous female mice. Furthermore, we used a pharmacological approach to manipulate adenosine receptor signaling in an attempt to reverse BDNF signaling deficits in synaptic plasticity in this disease-relevant RTT mouse model.

## 2. Results

### 2.1. The Magnitude of Hippocampal LTP Is Maintained in Mecp2^+/−^ Female Mice but BDNF Loses Its Effect on LTP Magnitude 

From previous works, it is known that LTP is impaired in *Mecp2*^−/y^ animals [15], and there is a strong impairment in the facilitation effect of BDNF on LTP [19,24]. However, data on alterations in LTP and BDNF signaling in *Mecp2*^+/−^ female animals are scarcer [29,34,35]. Therefore, we set out to investigate alterations in basal LTP and the effect of BDNF on this form of synaptic plasticity in a relevant RTT female mouse model.

The LTP magnitude in hippocampal slices from *Mecp2*^+*/−*^ female mice was not significantly different from that observed in hippocampal slices from age-matched WT animals (WT-LTP_CTR_ = 28.3 ± 4.2%, *n* = 6 and *Mecp2*^+/−^-LTP_CTR_ = 39.1 ± 5.8%, *n* = 8; *p* = 0.2, unpaired *t*-test; Figure 1C–E). Importantly, and despite the lack of major impairments in LTP in heterozygous females, there was a clear impairment in the actions of BDNF on LTP. Indeed, incubation with BDNF (20 ng/mL), as expected, enhanced LTP in WT slices (Figure 1C–E) but was unable to further facilitate LTP in *Mecp2*^+*/−*^ slices (WT-LTP_BDNF_ = 52.0 ± 8.2%, *n* = 6, paired Student’s *t*-test; *p* = 0.020; Figure 1C,E; and *Mecp2*^+/−^-LTP_BDNF_ = 41.8 ± 5.5%, *n* = 9; *p* = 0.500, paired Student’s *t*-test; Figure 1D,E). Post-tetanic potentiation (PTP) represents a short form of synaptic plasticity, resulting in an increased neurotransmitter release after a brief, high-frequency train of action potentials, was also analyzed [16]. BDNF (20 ng/mL) was able to significantly increase PTP in *Mecp2*^+/−^ female mice but not in WT animals (*Mecp2*^+/−^-PTP_CTR_ = 66.0 ± 7.6% and PTP_BDNF_ = 131.3 ± 19.0%, *n* = 7; *p* = 0.009; Figure 1D,F). 

Input/output (I/O) curves were performed to evaluate synaptic efficiency in the hippocampus of *Mecp2*^+/−^ mice (Figure 1G,H). Hippocampal slices from *Mecp2*^+/−^ mice display higher Emax levels compared with WT preparations (E_max-WT_ = 1.12 ± 0.09, *n* = 8 vs. E_max-*Mecp2*+/−_ = 1.86 ± 0.21, *n* = 8; *p* = 0.005, unpaired Student’s *t*-test; Figure 1G,H).

### 2.2. BDNF Protein Levels Are Decreased in the Hippocampus and Cortex of Symtomatic Heterozygous Mecp2^+/−^ Female Mice 

BDNF protein levels were characterized in the cortex and hippocampus of *Mecp2*^+/−^ female mice by Western blot alongside Mecp2 levels.

We found that BDNF protein levels were decreased both in the hippocampus and cortex of *Mecp2*^+/−^ when compared with aged-matched WT animals (HIP_WT_ = 100.0 ± 11.31%, *n* = 7 and HIP*_Mecp2_*_+/−_ = 64.6 ± 8.02, *n* = 5; *p* = 0.03; CTX_WT_ = 100.0 ± 12.26%, *n* = 13 and CTX*_Mecp2_*_+/−_ = 63.2 ± 10.09, *n* = 13; *p* = 0.04; unpaired *t*-test; Figure 2A,C,E). As expected, we also observed that Mecp2 protein levels were significantly decreased in the hippocampus of *Mecp2*^+/−^ when compared with WT animals. This clear reduction in Mecp2 levels was not observed in cortex homogenates from *Mecp2*^+/−^ when compared with WT animals (HIP_WT_ = 100.00 ± 12.80%, *n* = 7 vs. HIP*_Mecp2_*_+/−_ = 38.08 ± 4.31, *n* = 6; *p* = 0.001; CTX_WT_ = 100.00 ± 12.98%, *n* = 9 vs. CTX*_Mecp2_*_+/−_ = 74.3 ± 6.40, *n* = 8; *p* = 0.110; unpaired Student’s *t*-test; Figure 2B,D,E). This heterogeneity may reflect heterozygosity and consequent genetic mosaicism among *Mecp2*^+/−^ female mice.

### 2.3. TrkB-FL Protein Levels Are Not Changed in the Hippocampus of Mecp2^+/−^ Mice

Since differences were found in BDNF protein levels, we next investigated if TrkB receptor protein levels were also affected in the hippocampus and cortex of this RTT female mouse model.

In the brain of heterozygous *Mecp2*^+*/−*^ female mice, no significant differences were observed regarding TrkB-FL protein levels (HIP_WT_ = 100.00 ± 33.21%, *n* = 6 vs. HIP*_Mecp2_*_+/−_ = 66.60 ± 19.07, *n* = 6; *p* = 0.400; and CTX_WT_ = 100.00 ± 12.08%, *n* = 11 vs. CTX*_Mecp2_*_+/−_ = 109.20 ± 15.88, *n* = 11; *p* = 0.650; unpaired Student’s *t*-test; Figure 3A,C,E). No significant differences were detected in the levels of its truncated isoform, TrkB-Tc, either (HIP_WT_ = 100.00 ± 12.14%, *n* = 6 vs. HIP*_Mecp2_*_+/−_ = 103.50 ± 8.51, *n* = 5; *p* = 0.800; CTX_WT_ = 100.00 ± 11.06%, *n* = 10 vs. CTX*_Mecp2_*_+/−_ = 109.90 ± 10.95, *n* = 11; *p* = 0.160; unpaired Student’s *t*-test; Figure 3B,D,E).

### 2.4. Adenosine (Plus Inosine) Levels Remain Unaltered in the Cortex and Hippocampus of Heterozygous Mecp2^+/−^ Female Mice

The differences found in BDNF-TrkB-FL signaling could be aggravated by changes in the adenosinergic system, since adenosine has an important role in the modulation of BDNF signaling [36]. In this way, adenosine + inosine (an adenosine metabolite) amounts were measured in order to evaluate if adenosinergic system changes could also be involved in the aforementioned BDNF signaling deficits previously described.

Heterozygous *Mecp2*^+*/−*^ females had no significant (*p* > 0.05) differences in adenosine plus inosine content normalized to the wet tissue weight when compared with WT controls, both in the cortex (WT: 382 ± 71 nmol/g, *n* = 7 vs. *Mecp2*^+*/−*^: 382 ± 79 nmol, *n* = 8; Figure 4A) and in the hippocampus (WT: 225 ± 72 nmol, *n* = 5 vs. *Mecp2*^+*/−*^: 259 ± 94 nmol, *n* = 6; Figure 4B). Interestingly, cortical levels of adenosine + inosine (ADO + INO) were higher (about threefold) than those detected in the hippocampus (Figure 4A,B).

As expected, no changes in the activity adenosine deaminase (ADA—the enzyme responsible for catalyzing the irreversible deamination of adenosine to inosine and [37]) activity, determined as INO/ADO ratios, were observed at any given sample when comparing WT and heterozygous *Mecp2*^+*/−*^ female mice, both in the cortex (WT: 0.367 ± 0.095 vs. *Mecp2*^+*/−*^: 0.305 ± 0.040) and in the hippocampus (WT: 0.463 ± 0.090 vs. *Mecp2*^+*/−*^: 0.574 ± 0.095) (Figure 4C,D, respectively).

### 2.5. Mecp2^+/−^ Females Display Alterations in Hippocampal A_2A_R and A_1_R Adenosine Receptor Levels

We also addressed putative changes in A_1_R and A_2A_R adenosine receptor levels in hippocampal homogenates of Mecp2^+/−^ and WT female animals.

In summary, when comparing *Mecp2*^+/−^ to WT females, we observed (1) a significant increase in A_1_R protein levels in *Mecp2*^+/−^ females (HIP_WT_ = 100.00 ± 3.89%, *n* = 5 vs. HIP_Mecp2+/−_ = 116.60 ± 3.62%, *n* = 5, *p* = 0.01, unpaired Student’s *t*-test; Figure 5A,C) and (2) a significant decrease (~47%) in the levels of A_2A_R protein in *Mecp2*^+/−^ (HIP_WT_ = 100.00 ± 17.78%, *n* = 6 vs. HIP_Mecp2+/−_ = 52.47 ± 8.54%, *n* = 6, *p* = 0.04, unpaired Student’s *t*-test; Figure 5B,C).

### 2.6. Exogenous Activation of A_2A_R Restores BDNF-Induced LTP Facilitation in the Hippocampus of Heterozygous Mecp2^+/−^ Female Mice

Adenosine A_2A_R activation is known to boost TrkB-FL receptor signaling [36]. Therefore, we aimed to test if exogenous activation of A_2A_R with a potent specific agonist could compensate for the decreased A_2A_R levels found in the hippocampus of *Mecp2*^+/−^ female mice. In this way, we tested the effect of CGS21680 (10 nM) on LTP in hippocampal slices in the presence or absence of exogenously added BDNF (20 ng/mL) (see e.g., [38]).

Incubation with CGS21680 (10 nM) alone did not significantly change the magnitude of LTP in hippocampal slices from *Mecp2*^+/−^ mice (LTP_CTR_ = 32.74 ± 5.73, *n* = 5 vs. LTP_CGS_ = 41.27 ± 7.44, *n* = 5, *p* > 0.05; paired Student’s *t*-test; Figure 6A,C), though a tendency for an increase in LTP could be envisaged. Importantly, when CGS21680 (10 nM) was applied together with BDNF (20 ng/mL) to *Mecp2*^+/−^ hippocampal slices, the magnitude of LTP was significantly increased (LTP_CTR_ = 32.60 ± 9.74, *n* = 6 and LTP_CGS + BDNF_ = 45.36 ± 5.92, *n* = 6; paired *t*-test; Figure 6B,C). These data suggest that the facilitatory action of BDNF upon LTP in the hippocampus of *Mecp2*^+/−^ female mice can be rescued if both signaling pathways are simultaneously activated.

## 3. Discussion

Considering the genetic profile of RTT, heterozygous female mice are more representative of the genetic phenomena occurring in this syndrome when compared with *Mecp2*-null male mice [29,33]. Some studies suggest that, as a consequence of XCI and differences in skewing of the mutant X chromosome, among other emerging factors, patients carrying similar mutations might reveal differences in the clinical severity of the manifestation of the syndrome [39,40,41]. Despite this knowledge, the advantages associated with studying *Mecp2*-null mice, like high reproducibility, good characterization, and rapid establishment of symptomatology that mimics a great part of the symptoms present in patients [28,42], still turn female heterozygous mice into a backup model for RTT studies. The data presented on changes in BDNF signaling and in the adenosinergic system show, however, that some characteristics can cut across different phenotypic severities caused by total or partial loss of Mecp2 expression and/or function.

Even though regulation of the *Bdnf* gene by Mecp2 has been widely studied, the exact mechanisms behind this regulation are not clearly known [9,32]. Most of the studies describe a decrease in BDNF protein levels in *Mecp2*-null male mice, detectable at the onset of the first behavioral impairments [16]. Several studies demonstrate a significant decrease in BDNF levels in symptomatic *Mecp2*-null mice, not only in the cortex and hippocampus but also in other brain regions [15,16,17,18]. The decrease in BDNF levels shows a progression from the caudal brain regions affecting all brains around 7 weeks of age [17,43]. Studies in heterozygous female mice are scarce, but there is evidence for decreased levels of BDNF in the medulla, pons, sensory nodose ganglion, and cortex in symptomatic animals [35].

In the present work, the data obtained using *Mecp2* heterozygous females, *Mecp2*^+/−^, show a decrease in BDNF protein levels, both in the cortex and hippocampus of symptomatic animals, with a simultaneous loss of the facilitatory effect of BDNF upon LTP in the hippocampus. These results are in line with the previously mentioned studies showing a significate decrease in BDNF levels not only in the cortex and hippocampus of *Mecp2*^−/y^ males but also in other brain regions of symptomatic male *Mecp2*^−/y^ and *Mecp*2^+/−^ female models of RTT [15,16,17,18]

In contrast with the observed reduction in BDNF levels, no alteration was found in TrkB-FL levels in the hippocampus and cortex of *Mecp*2^+/−^ females. These two brain regions are highly affected in *Mecp2*^−/y^ animals and are associated with multiple functions altered in RTT, namely cognition [15,16,44]. Our results in *Mecp*2^+/−^ females differ from those previously obtained in *Mecp2*^−/y^ males, where a reduction in TrkB-FL in the cortex and hippocampus could potentially explain the inability of exogenous BDNF to potentiate synaptic plasticity. These arguments could also be used to justify the inefficacy of some therapeutic strategies grounded in the increase in BDNF levels [9,15].

Contrary to what has been observed in the hippocampus of *Mecp2*^−/y^ animals [13,37], heterozygous females do not show alterations in basal LTP. However, a recent study already described alterations in the hippocampal LTP of a symptomatic RTT female mouse model, heterozygous for *Mecp2* (*Mecp2*^tm1.1Jae^) [34]. Although these changes were described in an *Mecp2*-null model, the age and the strain differ from the one studied in the present study, suggesting that LTP impairments in *Mecp2* heterozygous models might also be dependent on the age and strain of the model used. Nevertheless, it is also important to take into account that different protocols could be used to study LTP. Indeed, the mentioned study used a stronger θ-burst protocol, which could also explain the discrepancy between both studies. Moreover, we found that the exogenous BDNF was unable to induce any potentiation of hippocampal LTP, as previously observed in *Mecp2*^−/y^ animals [15]. Importantly, the analysis of I/O curves demonstrated that deficits in the facilitation of hippocampal LTP were not due to alterations in synaptic efficacy but rather probably due to changes in BDNF signaling. Interestingly, our data showed a BDNF effect on PTP facilitation in *Mecp2*^+/−^ female animals. These data suggest that the BDNF facilitatory effect was transient and, therefore, insufficient to be translated into LTP in *Mecp2*^+/−^ female animals, in opposition to WT animals, where the BDNF faciliatory effect is just observed in long-term plasticity forms. In fact, diminished PTP is described in one study performed in *Mecp2*^−/y^ male animals [45]. The few characterizations of short-term forms of plasticity made, so far, in *Mecp2*-null mice models could be a missing piece that is important to understand if any mechanisms during short-term neuronal plasticity are changed and that could justify the alterations described in long-term plasticity forms in RTT [34,45,46,47].

Still, although not entirely superimposed to the alterations found in previous studies in *Mecp2*^−/y^ animals, in the model now studied, which displays a milder symptomatic phenotype, the dysregulation of BDNF signaling was also observed.

In addition, a dysregulation of the adenosinergic system in an RTT female mouse model was also presented in this study. Adenosine and BDNF signaling are tightly linked, since A_2A_R activation is known to increase BDNF and TrkB-FL levels; promote transactivation of TrkB-FL receptors and downstream signaling through the Gs-cAMP-PKA pathway; and promote translocation of TrkB-FL levels to lipid rafts in the cell membrane [21,24,36,48]. Therefore, deciphering the dysregulation in this system could bring new insights into the mechanisms contributing to BDNF signaling impairments in RTT.

Here, we observed alterations in the expression of adenosine receptors subtypes A_1_ and A_2A_, in which downstream signaling pathways play different roles in cell physiology, as A_1_R and A_2A_R are canonically coupled to Gi and Gs, respectively [49]. In the hippocampus of *Mecp2*^+/−^ female, we observed an increase in A_1_R and a decrease in A_2A_R protein levels. These results are in line with our previous study using *Mecp2*^−/y^ male animals, showing an increase in A_1_R and a reduction in A_2A_R levels both in the cortex and hippocampus [15]. The reduction in A_2A_R protein levels in the hippocampus of *Mecp2*^+/−^ female mice may, at least in part, be responsible for the lack facilitatory effect of BDNF upon LTP observed in this study.

However, in contrast with our observations in *Mecp2*^−/y^ male animals, where a reduction in overall adenosine levels was observed in the cortex and hippocampus, in the *Mecp2*^+/−^ female mouse model, adenosine levels remained unaltered [15]. This maintenance of the adenosine levels may contribute to a milder RTT phenotype compared with homozygous *Mecp2*^−/y^ males.

The mechanism underlying the overall alterations in A_1_R and A_2A_R levels in both RTT models remains elusive. However, we hypothesize the overexcitability associated with this pathology could be a triggering factor for these changes which, at some extent, could compensate for the excitatory–inhibitory imbalance, mainly described in *Mecp2*-null mice [44]. Remarkably, although lower levels of A_2A_R were detected, the available receptors were enough to respond to exogenous activation with a selective agonist, since CGS21680 was able to rescue the action of BDNF on LTP.

In this context, we observed that pharmacological activation of A_2A_R allowed the recovery of the facilitatory action of BDNF on LTP. These changes were congruent with those described previously in *Mecp2*^−/y^ mice, suggesting that the dysregulation of BDNF signaling and the adenosinergic system are a hallmark of RTT and transversal to different mouse models presenting different disease severities. Furthermore, these data also support the importance of adenosinergic system regulation for effective BDNF functions [15].

Taken together, the results presented here reinforce the hypothesis that adenosine-enhancing therapies could be a useful strategy in RTT where there is a decrease in BDNF signaling together with impairments in the adenosinergic system.

## 4. Materials and Methods

### 4.1. Animals

The experiments were performed on female symptomatic adult mice, according to previous studies [29,50] (*Mecp2*^−/+^) (26 to 30 weeks old—symptomatic stage), B6.129P2(C)-*Mecp2*tm1.1Bird/J (*Mecp2* knockout, deletion of exons 3 and 4 of the *Mecp2* gene) [50], and in wild-type (WT) littermates that were used as control. The genotype of the animals was determined by polymerase chain reaction, as previously described [51].

The animals were housed in a 12 h light/dark cycle, with food and water provided ad libitum. Experiments involving animals were taken into careful consideration in order to reduce the number of animals sacrificed. All animals were handled according to the Portuguese law on Animal Care and European Union guidelines (86/609/EEC).

### 4.2. Ex Vivo Electrophysiological Recordings

**Hippocampus isolation and slice preparation**: The animals were first anesthetized with isoflurane (1,2-Propylenglycol 50% (*v/v*)) in an anesthesia chamber. When animals showed the first signs of anesthetic state, like reduction in respiratory rate and lack of reflexes, they were sacrificed by decapitation. In order to have access to the brain, the skull was exposed by cutting the skin at the top of the head, and then the brain was carefully removed and placed in ice-cold artificial cerebrospinal fluid (aCSF—Krebs’ solution) containing 124 mM NaCl; 3 mM KCl; 1.25 mM NaH_2_PO_4_, 26 mM NaHCO_3_, 1 mM MgSO_4_, 2 mM CaCl_2_, and 10 mM glucose previously gassed with 95% O_2_ and 5% CO_2_, pH 7.4. The two brain hemispheres were separated through the midline, and the hippocampus was isolated, taking care not to damage it with the spatulas. When isolated, the hippocampus was cut perpendicularly to the long axis into slices with 400 µm thickness with the Mcllwain tissue chopper. Slices were then placed in a resting chamber in Krebs’ solution and permanently oxygenated at room temperature for one hour in order to recover [15].

**LTP induction and quantification**: After functional and energetic recovery, slices were transferred to the recording chamber for submerged slices, being continuously superfused with bathing solution (Krebs’ solution) gassed with 95% O_2_ and 5% CO_2_ at 32 °C. The flux of bathing solution was established at 3 mL/min and the drugs used were added to this superfusion solution. Recordings were obtained with an Axoclamp 2B amplifier and digitized (Axon Instruments, Foster City, CA, USA). Individual responses were monitored, and averages of eight consecutive responses were continuously stored on a personal computer with the LTP program [52]. Field excitatory postsynaptic potentials (fEPSPs) were recorded through an extracellular microelectrode (2–8 MW resistance, Harvard apparatus LTD, Fircroft way, Edenbridge, Kent) placed in the *stratum radiatum* of the CA1 area. Stimulation (rectangular 0.1 ms pulses, once every 10 s) was delivered through a concentric electrode placed on the Schaffer collateral–commissural fibers in the *stratum radiatum* near CA3-CA1 border. The stimulus intensity (80–260 µA) was initially adjusted to obtain a large fEPSP with a minimum population spike contamination. Stimulation was delivered alternatively to two independent pathways (previously tested by paired pulsed facilitation protocol). LTP was induced by a θ-burst protocol consisting of three trains of 100 Hz, three stimuli, separated by 200 ms, as previously described [19]. LTP was quantified as the % change in the average slope of the fEPSP taken 50 to 60 min after LTP induction in relation to the average slope of the fEPSP, measured during the 10 min that preceded the induction of LTP. In each individual experiment, the same LTP-inducing paradigm was delivered to each pathway. At 1 h after LTP induction, in one of the pathways (S1—Figure 1B), BDNF (20 ng/mL), CGS21680 (10 nM), or both were added to the superfusion solution, and LTP was induced in the second pathway (S2—Figure 1B) no less than 15 min after the drug perfusion and only after stability of fEPSP slope values was observed for at least 10 min. The effect of the drug tested upon LTP was evaluated by comparing the magnitude of LTP in the first pathway in the absence of BDNF (control pathway) with the magnitude of LTP in the second pathway in the presence of the drug (test pathway); each pathway was used as control or test on alternate days [15].

**Input–output curve (I/O):** After obtaining a stable baseline for at least 10 min, the stimulus delivered to the slice was decreased until no fEPSP was evoked and subsequently increased in 20 μA steps. For each stimulation intensity, data from three consecutive averages of 8 fEPSP were collected. Inputs delivered to slices typically ranged from 60 μA to a supramaximal stimulation of 320 μA. The input–output curve was plotted as the relationship of fEPSP slope vs. stimulus intensity, which provides a measure of synaptic efficiency, as previously described [53].

**Drugs:** BDNF was generously provided by Regeneron Pharmaceuticals; 2-[p-(2-carboxyethyl)phenethylamino]-50-N-ethylcarboxamido adenosine (CGS21680) was purchased from Sigma (Kawasaki, Japan). The maximum DMSO concentration (0.02%) applied to the preparations was devoid of action on fEPSPs (Tsvyetlynska et al., n.d.). Aliquots of the stock solutions were kept frozen at −20 °C until use.

### 4.3. Extraction and Analysis of Adenosine and Inosine by Liquid Chromatography with Diode Array Detection (HPLC/DAD)

Adenine nucleosides (adenosine and inosine) extracted from the cortex and hippocampus of WT and *Mecp^2^*^−/y^ female mice were measured by high-performance liquid chromatography with diode array detection (HPLC/DAD), as previously described [15]. Snap-frozen cortex and hippocampal tissue samples were stored at −80 °C until use. For extraction, the samples were defrosted (250 μL) in round-bottom microcentrifuge tubes, followed by thorough tissue homogenization using a mixture of ice-cold acetonitrile: methanol: water (1:2:2) solution containing 2-chloro-adenosine (5 μM) as internal standard; the obtained mixture was centrifuged at 16,000× *g* for 20 min at 4 °C. Tissue homogenization and centrifugation were repeated twice. The two recovered supernatant extracts (~250 μL each) were mixed together and then centrifuged again at 16,000× *g* (for 20 min at 4 °C) using a 50-kDa cutoff filter (Amicon Ultra-0.5 50 K Filter Device; Merck KGaA, Darmstadt, Germany). After filtration, supernatant extracts were divided into 15 μL aliquots and stored at −80 °C until analysis. Using this procedure, recovery of adenine nucleosides was higher than 95%, as determined by adding 2-chloro-adenosine (5 μM) as internal standard before extraction (see e.g., ref. [15]).

Extraction media containing purines were 1/10 diluted with ultrapure water before HPLC/DAD analysis. The chromatographic separation of nucleosides was carried out using an elution gradient composed of ammonium acetate (5 mM, with a pH of 6 adjusted with acetic acid) and methanol [54,55,56] (see Appendix A). Separation of adenosine and inosine was achieved by reversed-phase liquid chromatography through a Hypersil GOLD C18 column (5 μM, 2.1 mm × 150 mm) equipped with a guard column (5 μm, 2.1 mm × 1 mm) and assayed using a Finigan Thermo Fisher Scientific System LC/DAD, equipped with an Accela Pump (version 1.04.0016) coupled to an Accela Autosample (version 2.2.1), a diode array detector, and an Accela PDA (version 2.3.0) running the Xcalibur software (version 2.1.0.1139) chromatography manager (RRID:SCR_014593;Thermo Fisher Scientific Inc., Thousand Oaks, CA, USA). During the procedure, the flow rate was set at 200 μL·min^−1^, and the column temperature was maintained at 20 °C. The autosampler was set at 4 °C, and 50 μL of standard or sample was injected, in duplicate, for each HPLC analysis. Quantification of adenine nucleosides was carried out using calibration curves made of high-purity external standards, namely adenosine and inosine.

### 4.4. Western Blot

**Protein Extraction:** The respective brain areas intended for study were first dissected in ice-cold artificial cerebrospinal fluid (aCSF) solution: NaCl 124 mM; KCl 3 mM; NaH_2_PO_4_ 1.25 mM; NaHCO_3_ 26 mM; MgSO_4_ 1 mM; CaCl_2_ 2 mM; and glucose 10 mM, previously gassed with 95% O_2_ and 5% CO_2_, pH 7.4 solution, washed in PBS solution (NaCl 137 mM, KCl 2.1 mM, KH_2_PO_4_ 1.8 mM and Na_2_HPO_4_·2H_2_O 10 mM, pH 7.4), and immediately snap-frozen and stored at −80 °C until homogenates preparation. Snap-frozen brain samples were disrupted using a sonicator (Sonic & Materials Inc., Newtown, CT, USA) in 250 μL (hippocampus and striatum) or 500 μL (cerebellum, cortex and brainstem) of Ristocetin Induced Platelet Agglutination (RIPA) lysis buffer (1 M Tris pH 8.0, 0.5 M EDTA pH 8.0, 5 M NaCl, 0.1% Sodium Dodecyl Sulfate (SDS), 10% Nonidet P-40 (NP-40), 50% Glycerol) supplemented with cOmpleteTM Mini protease-inhibitor cocktail tablets (Mini-Complete EDTA-free; Roche Applied Science, Penzberg, Germany). All lysates were then vortexed and sonicated (3 cycles of 15 s). The protein content in the supernatant was determined by Bio-Rad DC reagent assay (Bio-Rad, Hercules, CA, USA) [15].

**Protein Electrophoresis, Transfer, and Detection/Quantification:** Equal amounts of protein were loaded (70 μg except for A_2A_R blot: 200 μg) and separated on 10–12% sodium dodecyl sulfate–polyacrylamide gel electrophoresis (SDS-PAGE) and then transferred to polyvinylidene fluoride (PVDF) membrane (GE Healthcare, Chicago, IL, USA). To check protein transfer efficiency, membranes were stained with Ponceau S solution. After blocking with a 5% nonfat dry milk solution in TBS-T (20 mM Tris base, 137 mM NaCl and 0.1% Tween-20), membranes were washed three times with TBS-T, before incubation with the primary antibody, diluted in 3% BSA solution in TBS-T (overnight at 4 °C) and secondary antibodies (1:10,000, Bio-Rad, Hercules, CA, USA) diluted in blocking solution (1 h at room temperature). Immunoreactivity was visualized using an ECL chemiluminescence detection system (Amersham-ECL Western Blotting Detection Reagents from GE Healthcare), and band intensities were quantified by digital densitometry (ImageJ 1.45 software). GAPDH (1:5000, #AM4300, Ambion by life technologies, Carlsbad, CA, USA) bands were used as loading control. The primary antibodies used were Anti-A_2A_R from Merk Millipore (1:1500, 05-717, Darmstadt, Germany), Anti-A_1_R from Santa Cruz Biotecnology (1:1000, sc-28995, Dallas, TX, USA), Anti-TrkB from BD Transduction Laboratories (1:1500, 610101, Franklin Lakes, NJ, USA), and Anti-BDNF from Abcam (1:1500, ab108319, Cambridge, UK) and Anti-Mecp2 (1:1500, ab50005, Cambridge, UK) [15].

### 4.5. Data Analysis

The data are expressed as mean ± SEM of the n number of independent experiments. The significance of differences between the means of 2 conditions was evaluated by paired or unpaired *t*-tests; Welch correction was used in unpaired *t*-test as appropriate. Nonlinear regressions were used to fit data pertaining to I/O curves and radioligand binding experiments. Values of *p* < 0.05 were considered to represent statistically significant differences. GraphPad Prism 5.00 was used to perform statistical analysis.

## Figures and Tables

**Figure 1 ijms-24-16249-f001:**
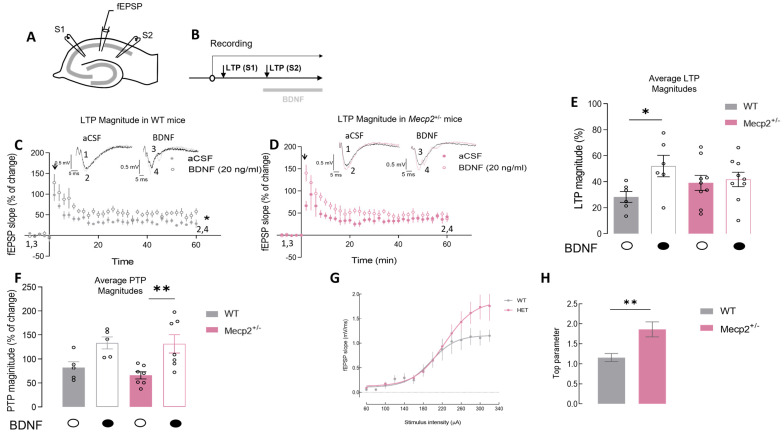
The magnitude of hippocampal LTP is maintained in *Mecp2*^+/−^ female mice, but BDNF loses the facilitatory effect upon LTP in *Mecp2*^+/−^ female animals: (**A**) Schematic representation of a transverse hippocampal slice and the recording configuration used in the present study. (**B**) Schematic representation of the experimental protocol used to induce LTP and to test the effect of BDNF at a concentration of 20 ng/mL (for a detailed description, see Section 4.2). Panels (**C**,**D**) represent the time course of averaged fEPSP slopes induced after θ-burst stimulations in hippocampal slices isolated from WT (grey symbols, *n* = 6) or *Mecp2*^+/−^ (pink symbols, *n* = 8) mice, either in the absence (dark-filled circles bellow the bar, light bars) or in the presence (white-filled circles bellow the bar, dark bars) of BDNF. Recording traces from representative experiments are shown for WT (black trace-basal synaptic potential (1,3); gray trace-synaptic potential after LTP (2,4)) and *Mecp2*^+/−^ (pink) animals on top of each graph (1 and 3—baseline; 2 and 4—LTP, 1 h after the θ-burst). The arrow indicates LTP induction. In (**E**), the histogram shows the magnitude of LTP in the absence (dark—LTP S1) and the presence of BDNF (20 ng/mL, light bars—LTP S2) obtained in hippocampal slices from WT (gray) or *Mecp2*^+/−^ (pink) animals; the PTP magnitude is shown in (**F**). Panels (**G**,**H**) show the input/output (I/O) curves and the respective means of the top parameter corresponding to responses generated by various stimulation intensities (60–340 μA) in WT (gray circles, *n* = 8) and *Mecp2*^+/−^ (pink circles, *n* = 8) mice hippocampal slices. * *p* < 0.05; ** *p* < 0.01 (paired Student’s *t*-test) as compared with the absence of BDNF in the same experiments.

**Figure 2 ijms-24-16249-f002:**
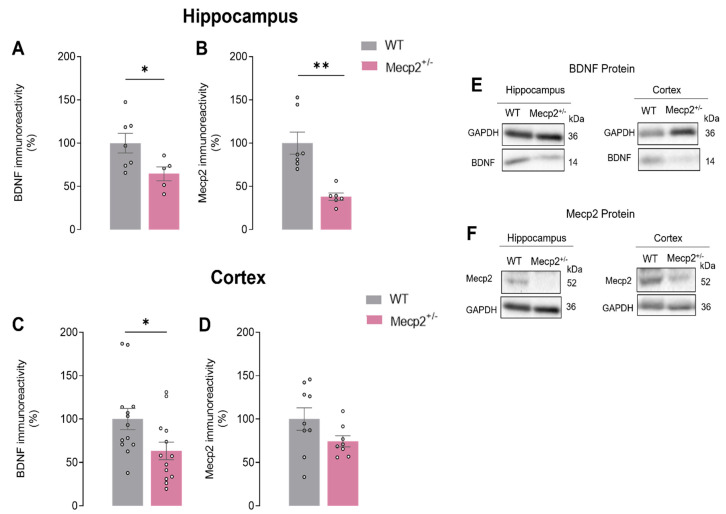
BDNF and Mecp2 protein levels in *Mecp2*^+/−^ animals: In (**A**,**C**), the averaged BDNF density levels are shown (WT_hip_, *n* = 7; WT_ctx_, *n* = 13; *Mecp2*^+/−^_hip_, *n* = 5; *Mecp2*^+/−^_ctx_, *n* = 13), while (**B**,**D**) show Mecp2 (WT_hip,_ *n* = 7; WT_ctx_, *n* = 9; *Mecp2*^+/−^_hip_, *n* = 6; *Mecp2*^+/−^_ctx_, *n* = 8) density levels evaluated by Western blot analysis in hippocampal and cortical homogenates from WT (gray) and *Mecp2*^+/−^ (pink) animals at 26–30 weeks of age. (**E**,**F**) show the representative bands obtained. Immunoreactivity against GAPDH was used for normalization purposes. All values are mean ± standard error of mean (SEM). * *p* < 0.05; ** *p* < 0.01 (unpaired Student’s *t*-test) against WT values. Each circle represents one independent *n*.

**Figure 3 ijms-24-16249-f003:**
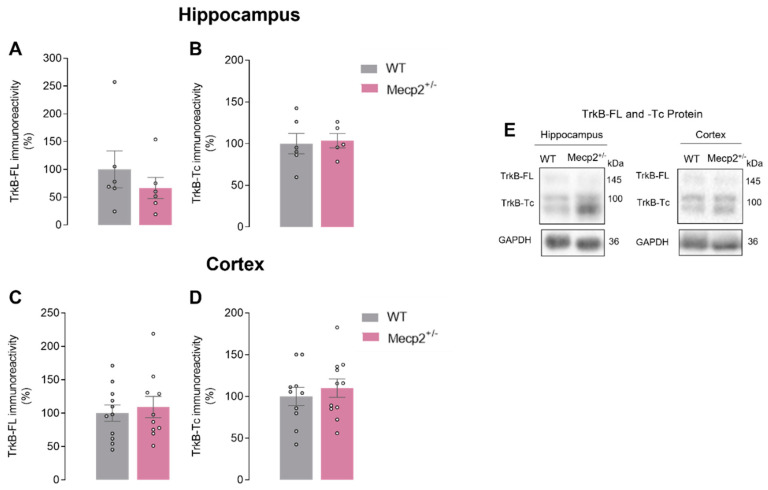
TrkB-FL and TrkB-Tc protein levels in heterozygous *Mecp2*^+/−^ female mice: In (**A**,**C**), averages of TrkB-FL levels density are shown (WT_hip_, *n* = 6; WT_ctx_, *n* = 11; *Mecp2*^+/−^_hip_, *n* = 6; *Mecp2*^+/−^_ctx_, *n* = 11). In (**B**,**D**), TrkB-Tc (WT_hip_, *n* = 6; WT_ctx_, *n* = 10; *Mecp2*^+/−^_hip_, *n* = 5; *Mecp2*^+/−^_ctx_, *n* = 11) levels are shown by Western blot analysis in hippocampal and cortical homogenates from WT (gray) and *Mecp2*^+/−^ (pink) animals at 26–30 weeks old. Representative immunoblots are shown in (**E**). Immunoreactivity against GAPDH was used for normalization purposes. All values are mean ± SEM. Unpaired Student’s *t*-test against WT values. Each circle represents one independent *n*.

**Figure 4 ijms-24-16249-f004:**
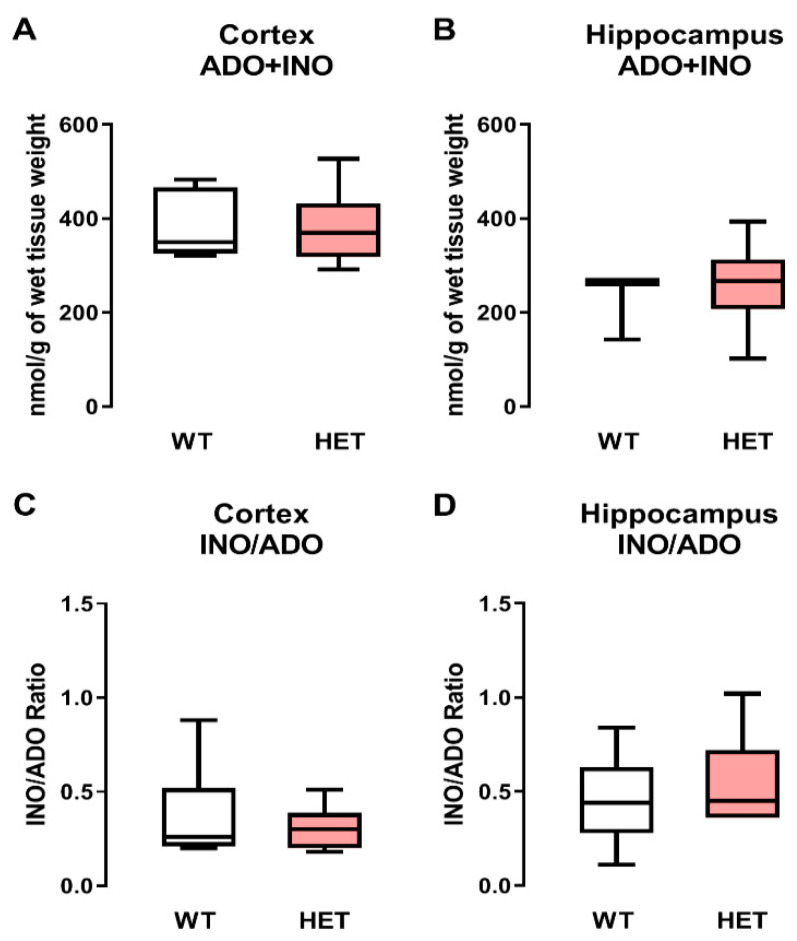
Adenosine (ADO) plus inosine (INO) content in the cortex and hippocampus of WT and heterozygous *Mecp2*^+/−^ female mice: Panels (**A**,**B**) represent the amount of ADO+INO in nmol/g of wet tissue weight detected by HPLC/DAD from extracts of the cortex and hippocampus, respectively, of wild-type (WT, gray bars) and *Mecp2*^+/−^ (HET, pink bars) female mice. Panels (**C**,**D**) represent the adenosine deaminase (ADA) activity given by the INO/ADO ratio in extracts of the cortex and hippocampus, respectively, of WT (gray bars) and *Mecp2*^+/−^ (HET, pink bars) female mice. Box-and-whiskers plots are represented with whiskers ranging from minimum to maximum values; the horizontal lines inside boxes indicate the corresponding medians. Each data set represents five to eight individuals (see text for details); duplicate measurements were performed for each individual experiment.. Typical chromatograms obtained using these experimental settings are shown in Appendix A.

**Figure 5 ijms-24-16249-f005:**
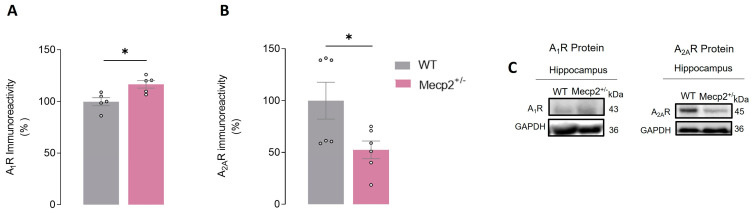
A_1_R and A_2A_R protein levels in the hippocampus of *Mecp2*^+/−^ female mice: Histograms represent averages of A_1_R ((**A**), WT_hip_, *n* = 5 and Mecp2^+/−^_hip_, *n* = 5) and A_2A_R ((**B**), WT_hip_, *n* = 6 and Mecp2^+/−^_hip_, *n* = 6) immunoreactivity in hippocampal homogenates from WT (gray bars) and *Mecp2*^+/−^ (pink bars) female mice at 26–30 weeks-old. In (**C**), representative immunoblots used for quantification are shown. The immunoreactivity against GAPDH was used for normalization purposes. All values are mean± SEM. * *p* < 0.05 (unpaired Student’s *t*-test) against WT values. Each circle represents one independent *n*.

**Figure 6 ijms-24-16249-f006:**
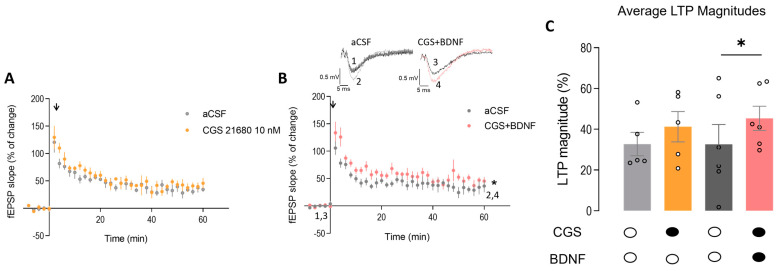
Exogenous activation of A_2A_R restores the facilitatory effect of BDNF on hippocampal LTP in heterozygous *Mecp2*^+/−^ female mice: Panels (**A**,**B**) show the time course of the averaged field excitatory postsynaptic potential (fEPSP) slope after a θ-burst stimulation of hippocampal slices from Mecp2^+/−^ female mice obtained in the absence (gray circles) and in the presence (orange circles) of the selective A_2A_R agonist, CGS21680 (10 nM), alone ((**A**), *n* = 5) or coapplied with BDNF (20 ng/mL, (**B)**, pink circles, *n* = 6). The A_2A_R agonist was applied 60 min after induction of LTP in the first pathway (gray circles—control LTP) and at least 15 min before induction of LTP in the second pathway (orange, pink circles). Representative traces from representative experiments are shown (black trace—basal synaptic potential (1,3); pink trace—synaptic potential after LTP (2,4)). The bar graph in panel (**C**) shows the percentage changes in the slope of fEPSP computed from panels (**A**,**B**) where dark-filled circles bellow the bars represent the presence of the drug and white-filled circles the absence of the drug. Gray bars represent control LTP (light gray—control experience for CGS alone, dark gray—control experience for CGS + BDNF experience), orange bar represents CGS alone effect upon LTP and pink bar CGS + BDNF effect upon LTP. Data are mean ± SEM. * *p* < 0.05 (paired Student’s *t*-test). Each circle on bars represents o one independent *n*.

## Data Availability

The data presented in this study are available on request from the corresponding author. The data are not publicly available due to ongoing studies related with this project.

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
