# Peer review of "Adenosinergic System and BDNF Signaling Changes as a Cross-Sectional Feature of RTT: Characterization of Mecp2 Heterozygous Mouse Females"

_ijms, 2023, doi:10.3390/ijms242216249_

Round 1
Reviewer 1 Report
Comments and Suggestions for Authors
Ms. ID # ijms-2654762
The AAs investigated, for the first time, the BDNF and adenosine signaling in a heterozygous female mouse model of Rett syndrome (Mecp2 +/-), an X-linked neurodevelopmental disease mainly caused by mutations of MECP2 gene. In a previous study, the same AAs observed that, in a Mecp2 knockout (Mecp2−/y) mouse model, BDNF signaling is impaired together with compromission of adenosinergic system.
In the present study, the AAs showed reduced BDNF protein levels in heterozygous female Mecp2 +/- mice as well as loss of BDNF-induced facilitation of hippocampal long-term potentiation (LTP), downregulated A2AR density, and increased A1R density in cortex and hippocampal sections. No variation was observed in adenosine and inosine levels in cortex and hippocampal samples from Mecp2 +/- mice. Furthermore, the AAs demonstrated that exogenous activation of A2AR restored the facilitatory effect of BDNF on hippocampal LTP in the experimental model of disease suggesting a new potential therapeutic strategy for Rett syndrome.
Unfortunately, the findings could be of interest but at this time the manuscript is not acceptable for publication.
A major revision is requested.
Main criticisms:
1) Article title and major criticism. The manuscript explored neither behavioural aspects nor disease severity of the female Mecp2 +/- mice as compared to wt mice. For this reason, in my opinion, the use of “..phenotype..” resulted inappropriate. If available, the AA should implement the results section with behavioural / disease severity scoring improving the significance of the findings.
Otherwise, the article title should be changed and this relevant limit should be reported in the manuscript.
2 2) Introduction. In order to provide an updated point-of-view of the topic, the AAs should update and cited some recent articles.
3 3) Methods. The AA should add the description of used statistical tests.
4) Methods. What is the method for adenine nucleoside analysis? LC/DAD or HPLC/DAD ?
5) Results. The AA should move some statements in the Discussion section (for example lines 127-131, 175-178). Moreover, the AA should avoid to mention their previous article at the start of each paragraph. The AA should simply describe the findings in the Mecp2+/- mouse model. All the other comparisons should be reported in the Discussion section.
Line 182. “(Figure 4A and 11B….)”. I did not find the Figure 11B. Please clarify.
6) Figure 3. The AA should explain what are the comparisons statistically significant. I did not found the asterisk. Please check.
7) Figure 4. The presentation should be improved. For example, in the panels A and B, the y-axis title should contain ADO+INO (nmol/…). Please specify the concentration unit. In the panels C and D the y-axis title should be INO/ADO ratio.
8) Discussion. This section should be implemented with updated literature and statements from the results section. The text should be re-organized
Minor criticisms:
-The text should be revised due to grammar and styling errors.
-Line 398 the symbol @ should be replaced with °
Comments on the Quality of English Language
Moderate editing of English language required
Author Response
Main criticisms:
1) Article title and major criticism. The manuscript explored neither behavioural aspects nor disease severity of the female Mecp2 +/- mice as compared to wt mice. For this reason, in my opinion, the use of “..phenotype..” resulted inappropriate. If available, the AA should implement the results section with behavioural / disease severity scoring improving the significance of the findings. Otherwise, the article title should be changed and this relevant limit should be reported in the manuscript.
Answer: We appreciate the reviews and all the criticisms raised. The suggested changes will certainly help to improve the quality of our manuscript. With regard to the first comment, we agree with the suggested changes. Thus, the title and the different sections have been modified, since the main objective of this study was to explore the molecular alterations in BDNF and the adenosinergic system, as well as their functional repercussions in LTP, in heterozygous females. Thus, the title has been changed to "Alterations in the adenosinergic system and BDNF signaling as a cross-sectional feature of RTT: characterization of Mecp2 heterozygous female mice".
2) Introduction. In order to provide an updated point-of-view of the topic, the AAs should update and cited some recent articles.
Answer: New updated references were added, as suggested, to better support bibliographically our manuscript. The new references are:
- Petriti, U.; Dudman, D.C.; Scosyrev, E.; Lopez-Leon, S. Global Prevalence of Rett Syndrome: Systematic Review and Meta-Analysis. Systematic Reviews 2023, 12, 5, doi:10.1186/s13643-023-02169-6.
- Gold, W.A.; Krishnarajy, R.; Ellaway, C.; Christodoulou, J. Rett Syndrome: A Genetic Update and Clinical Re-view Focusing on Comorbidities. ACS Chemical Neuroscience 2018, 9, 167–176.
- Sharifi, O.; Yasui, D.H. The Molecular Functions of MeCP2 in Rett Syndrome Pathology. Front Genet 2021, 12, 624290, doi:10.3389/fgene.2021.624290.
- KowiaÅ„ski, P.; Lietzau, G.; Czuba, E.; WaÅ›kow, M.; Steliga, A.; MoryÅ›, J. BDNF: A Key Factor with Multipotent Impact on Brain Signaling and Synaptic Plasticity. Cell Mol Neurobiol 2018, 38, 579–593, doi:10.1007/s10571-017-0510-4.
- Wang, H.; Chan, S.; Ogier, M.; Hellard, D.; Wang, Q.; Smith, C.; Katz, D.M. Dysregulation of Brain-Derived Neurotrophic Factor Expression and Neurosecretory Function in Mecp2 Null Mice. The Journal of neuroscience : the official journal of the Society for Neuroscience 2006, 26, 10911–10915, doi:10.1523/JNEUROSCI.1810-06.2006
- Ogier, M.; Wang, H.; Hong, E.; Wang, Q.; Greenberg, M.E.; Katz, D.M. Brain-Derived Neurotrophic Factor Ex-pression and Respiratory Function Improve after Ampakine Treatment in a Mouse Model of Rett Syndrome. J Neurosci 2007, 27, 10912–10917, doi:10.1523/JNEUROSCI.1869-07.2007.
- Bliss, T.V.; Collingridge, G.L. A Synaptic Model of Memory: Long-Term Potentiation in the Hippocampus. Nature 1993, 361, 31–39, doi:10.1038/361031a0.
- Martin, S.J.; Grimwood, P.D.; Morris, R.G. Synaptic Plasticity and Memory: An Evaluation of the Hypothesis. Annu Rev Neurosci 2000, 23, 649–711, doi:10.1146/annurev.neuro.23.1.649.
- Ribeiro, M.C.; MacDonald, J.L. Sex Differences in Mecp2-Mutant Rett Syndrome Model Mice and the Impact of Cellular Mosaicism on Phenotype Development. Brain Res 2020, 1729, 146644, doi:10.1016/j.brainres.2019.146644
- Good, K.V.; Vincent, J.B.; Ausió, J. MeCP2: The Genetic Driver of Rett Syndrome Epigenetics. Frontiers in Genet-ics 2021, 12.
3) Methods. The AA should add the description of used statistical tests.
Answer: Thank you for noting this. This section was mistakenly deleted in the first version of the manuscript. It has now been included in lines: 530-537.
4) Methods. What is the method for adenine nucleoside analysis? LC/DAD or HPLC/DAD ?
Answer: The method used for adenine nucleoside analysis was HPLC/DAD. It is now properly written in all sections where it is mentioned.
5) Results. The AA should move some statements in the Discussion section (for example lines 127-131, 175-178). Moreover, the AA should avoid to mention their previous article at the start of each paragraph. The AA should simply describe the findings in the Mecp2+/- mouse model. All the other comparisons should be reported in the Discussion section.
Answer: All result section was reviewed, as suggested, to be more focus on the newly described findings of the present manuscript.
- Line 182. “(Figure 4A and 11B….)”. I did not find the Figure 11B. Please clarify.
Answer: Thank you for the correction. This was a typing mistake. It was now corrected to Figure 4B (line:211).
6) Figure 3. The AA should explain what are the comparisons statistically significant. I did not found the asterisk. Please check.
Answer: Statistical comparisons of the results shown in figure 3 revealed no statistically significant differences. The legend has been changed and the asterisk removed for a better interpretation of these results.
7) Figure 4. The presentation should be improved. For example, in the panels A and B, the y-axis title should contain ADO+INO (nmol/…). Please specify the concentration unit. In the panels C and D the y-axis title should be INO/ADO ratio.
Answer: The figure 4 (panels A-D) was changed as suggested.
8) Discussion. This section should be implemented with updated literature and statements from the results section. The text should be re-organized
Answer: All this section was reviewed and literature updated.
The new references are:
- Cristalli, G.; Costanzi, S.; Lambertucci, C.; Lupidi, G.; Vittori, S.; Volpini, R.; Camaioni, E. Adenosine Deami-nase: Functional Implications and Different Classes of Inhibitors. Med Res Rev 2001, 21, 105–128, doi:10.1002/1098-1128(200103)21:2<105::aid-med1002>3.0.co;2-u.
- Xiol, C.; Vidal, S.; Pascual-Alonso, A.; Blasco, L.; Brandi, N.; Pacheco, P.; Gerotina, E.; O’Callaghan, M.; Pineda, M.; Armstrong, J.; et al. X Chromosome Inactivation Does Not Necessarily Determine the Severity of the Phe-notype in Rett Syndrome Patients. Sci Rep 2019, 9, 11983, doi:10.1038/s41598-019-48385-w.
- Borea, P.A.; Gessi, S.; Merighi, S.; Vincenzi, F.; Varani, K. Pharmacology of Adenosine Receptors: The State of the Art. Physiological Reviews 2018, 98, 1591–1625, doi:10.1152/physrev.00049.2017.
Minor criticisms:
-The text should be revised due to grammar and styling errors.
Answer: All text was reviewed.
-Line 398 the symbol @ should be replaced with °
Answer: Typing error. It was changed accordingly.
Reviewer 2 Report
Comments and Suggestions for Authors
The manuscript from Miranda-Lourenço and colleagues with the title: “The Rett syndrome phenotype of heterozygous Mecp2 female mice is associated with a milder impact on BDNF/Adenosine synergism” further analyzes the effect of a mecp2 knockout on BDNF signaling and the adenosinergic system known to act synergistically in the brain. Indeed both BDNF signaling and the adenosinergic system have already been shown to be altered in mecp2-null male mice. The current paper instead analyzes female mecp2 heterozygous mice (mecp2+/-) known to present a less severe phenotype and to better represent the majority of Rett syndrome (RTT) patients. From their results the authors conclude that: 1) while long term potentiation is not impaired in Mecp2+/- females compared to wildtype (WT), its magnitude in Mecp2+/- is not influenced by a BDNF treatment; 2) BDNF expression is lower than in WT both in the hippocampus and cortex of Mecp2+/-, whereas the levels of the full length TrkB receptor are not affected; 3) no differences could be observed in the adenosine levels between the two genotypes, but while the expression of the receptor A1R was increased and the one of A2AR was decreased in the hippocampus; 4) application of a AA2R agonist restores the BDNF-induced LTP facilitation in Mecp2+/- females.
The study of Mecp2+/- female mice is relevant to the clinical situation and addressing the BDNF signaling and the adrenergic system in this context it is indeed interesting for their possible value as therapeutic targets. However, the study presents a couple of issues that should be addressed before the manuscript can be accepted for publication.
Major points:
1) As far as I understand the method description, the LTP measurements were performed as follows: two independent pathways were stimulated in each slice. LTP was induced at the first pathway without any treatment, was recorded for one hour, then a treatment was performed and the second LTP was induced in the second pathway. So if I get it correctly the “control” curve in the graphs is always the result of the first LTP induction and the “BDNF” or “CGS or CGS+BDNF” curve is always the result of the second LTP induction. This should be better clarified also in the figure legends;
2) What the two pathways used for recording are is not really specified? From the position of the electrodes in the scheme in figure 1, I assume that they do not derive from two different axonal populations of the Shaffer collateral? This should be explained;
3) Performing the experiment in this way the effect of the treatment is always assessed on recordings that were performed in slices that are at least one hour older that when the first, control LTP was induced at the first pathway. I find this a strange experimental design since the decay of the slices might be different in different mouse genotypes. How is this approach much better than comparing control and treatment performed in different slices of the same mouse but with the same time course? I think that this should at least be described more in detail and discussed.
4) Regarding the LTP measurements, is the given n referring to the number of mice analyzed or the number of recorded slices? In this case from how many independent mice were the measurements performed. Both numbers should be given to be able to evaluate the solidity of the results and the statistical analysis;
5) Regarding figure 1: the authors conclude that the magnitude of LTP is not significantly different when comparing slices for mecp2+/- to WT slices and refer to figure 1C and D. However, in these two graphs the LTP magnitude of WT and mecp2+/- are not directly compared. Also it is not clear to which the unpaired t-test mentioned in the results for figure 1C and D refers to? Maybe to figure 1E? then however the correct test would be a two-way ANOVA, allowing to compare not only the effect of the treatment versus the control for each genotype but also the effect of genotype on the LTP magnitude.
6) Regarding the western blot images shown in figure2, 3 and 5, the loading controls of each blot should be added to the figures.
7) The method used to quantify the western blot analysis of protein expression should be describe in the methods with more details. How many loading and gel repetitions were used?
8) The authors write that the experiments were performed in symptomatic animals: does this refer to the age of the mice corresponding to the symptomatic stage of the diseases? Or were mice selected because of they showed specific symptoms? This should be clarified.
Minor points:
1) The genetic background of the mice should be mentioned
2) Some mistakes in the English writing should be corrected. E.g. line 33 “once” is not the right word in this context; line 296 “shown” should be “ showing”
3) Line 180: “gender” should be changed into “sex”. Indeed, gender is a socio-cultural concept and cannot be studied in biology.
Comments on the Quality of English Language
In few cases words are not correct for the context in which they are used. Please improve this
Author Response
1) As far as I understand the method description, the LTP measurements were performed as follows: two independent pathways were stimulated in each slice. LTP was induced at the first pathway without any treatment, was recorded for one hour, then a treatment was performed and the second LTP was induced in the second pathway. So if I get it correctly the “control” curve in the graphs is always the result of the first LTP induction and the “BDNF” or “CGS or CGS+BDNF” curve is always the result of the second LTP induction. This should be better clarified also in the figure legends;
Answer: We appreciate the reviews and all the criticisms raised. The suggested changes will certainly help to improve the quality of our manuscript.
Your interpretation is correct. For a better understanding of the used methodology, in the legend of Figure 1, we clarified which condition corresponds to each of the routes, via S1 and S2 (as shown in Figure 1B). It is important to clarify that the routes were alternated from experiment to experiment (the S1 route in one experiment became the S2 route in the next and vice versa, as explained in methods – section 4.2)
2) What the two pathways used for recording are is not really specified? From the position of the electrodes in the scheme in figure 1, I assume that they do not derive from two different axonal populations of the Shaffer collateral? This should be explained;
Answer: As mentioned in methods 4.2 (LTP induction and quantification) field excitatory post-synaptic potentials (fEPSPs) were recorded through an extracellular microelectrode placed in the stratum radiatum of the CA1 area. Stimulation was delivered alternatively to two independent pathways from two different axonal populations of Shaffer collateral. The independency of the pathways was tested by paired pulse facilitation protocol (this information is now better explained in methods – sections 4.2).
3) Performing the experiment in this way the effect of the treatment is always assessed on recordings that were performed in slices that are at least one hour older that when the first, control LTP was induced at the first pathway. I find this a strange experimental design since the decay of the slices might be different in different mouse genotypes. How is this approach much better than comparing control and treatment performed in different slices of the same mouse but with the same time course? I think that this should at least be described more in detail and discussed.
Answer: This is an important question. Using two different slices has the possible advantage that in the test condition (in the presence of drugs) it is not necessary to wait the time needed to record the first LTP (control condition). However, the use of different slices entails inherent disadvantages, since the slices can have different morphology and can be possibly taken from different areas of the hippocampus which can be a bias. The use of the same slice for both conditions (test and control) through the use of two stimulation pathways removes this bias. Although we calculate the magnitude of the control LTP 60 minutes after LTP induction, the fEPSP in this pathway are recorded until the end of the LTP recording in the second pathway, which allows us to control the decay of synaptic transmission or any other intercurrence.
4) Regarding the LTP measurements, is the given n referring to the number of mice analysed or the number of recorded slices? In this case from how many independent mice were the measurements performed. Both numbers should be given to be able to evaluate the solidity of the results and the statistical analysis;
Answer: In LTP measurements, n is referring to the number of mice analysed. For each mice, one slice was used. All n are referring to independent experiments. This information was now added methods, section 4.5.
5) Regarding figure 1: the authors conclude that the magnitude of LTP is not significantly different when comparing slices for mecp2+/- to WT slices and refer to figure 1C and D. However, in these two graphs the LTP magnitude of WT and mecp2+/- are not directly compared. Also it is not clear to which the unpaired t-test mentioned in the results for figure 1C and D refers to? Maybe to figure 1E? then however the correct test would be a two-way ANOVA, allowing to compare not only the effect of the treatment versus the control for each genotype but also the effect of genotype on the LTP magnitude.
Answer: We did not compare data obtain and represented in Figure 1C vs 1D. As explained, we just show, in each phenotype (WT and Mecp+/-), the BDNF effect upon basal LTP. In this way, we performed a paired t-test because no comparison was made between phenotypes. Figure 1E represents the magnitudes of all conditions but we just analysed the differences within each phenotype, not between them. All the analysis made are described in results section 2.1.
6) Regarding the western blot images shown in figure2, 3 and 5, the loading controls of each blot should be added to the figures.
Answer: The information regarding loading control was just missing in figure 2. We added the missed information.
7) The method used to quantify the western blot analysis of protein expression should be describe in the methods with more details. How many loading and gel repetitions were used?
Answer: Each n represents an independent experiment. For western-blot we performed one assay for our target protein with the respective loading control in the same gel.
8) The authors write that the experiments were performed in symptomatic animals: does this refer to the age of the mice corresponding to the symptomatic stage of the diseases? Or were mice selected because of they showed specific symptoms? This should be clarified.
Answer: According to The Jackson Laboratory (https://www.jax.org/strain/003890https://www.jax.org/strain/003890) symptomatic stage starts at 6 months of age. Some other studies observed, chronologically, the different stages of this model, confirming that at 26 weeks of age symptoms are present (Refs. 29,51). For a better clarification, we added this explanation in out methodology (4.1 section, line 406-407)
Minor points:
1) The genetic background of the mice should be mentioned.
Answer: The genetic background is stated in methodology section 4.1, (lines 392-394)
2) Some mistakes in the English writing should be corrected. E.g. line 33 “once” is not the right word in this context; line 296 “shown” should be “showing”.
Answer: We appreciated your revision. All text was reviewed and the mistakes corrected.
3) Line 180: “gender” should be changed into “sex”. Indeed, gender is a socio-cultural concept and cannot be studied in biology.
Answer: We agree with the observation made. Gender was replaced by sex.